# The Construction of a Multi-Gene Risk Model for Colon Cancer Prognosis and Drug Treatments Prediction

**DOI:** 10.3390/ijms25073954

**Published:** 2024-04-02

**Authors:** Liyang Gao, Ye Tian, Erfei Chen

**Affiliations:** 1Institute of Preventive Genomic Medicine, School of Life Sciences, Northwest University, Xi’an 710069, China; 2Key Laboratory of Resource Biology and Biotechnology in Western China, Ministry of Education, School of Life Sciences, Northwest University, Xi’an 710069, China

**Keywords:** colon cancer, differentially expressed genes (DEGs), survival analysis, LASSO regression analysis, tumor microenvironment

## Abstract

In clinical practice, colon cancer is a prevalent malignant tumor of the digestive system, characterized by a complex and progressive process involving multiple genes and molecular pathways. Historically, research efforts have primarily focused on investigating individual genes; however, our current study aims to explore the collective impact of multiple genes on colon cancer and to identify potential therapeutic targets associated with these genes. For this research, we acquired the gene expression profiles and RNA sequencing data of colon cancer from TCGA. Subsequently, we conducted differential gene expression analysis using R, followed by GO and KEGG pathway enrichment analyses. To construct a protein–protein interaction (PPI) network, we selected survival-related genes using the log-rank test and single-factor Cox regression analysis. Additionally, we performed LASSO regression analysis, immune infiltration analysis, mutation analysis, and cMAP analysis, as well as an investigation into ferroptosis. Our differential expression and survival analyses identified 47 hub genes, and subsequent LASSO regression analysis refined the focus to 23 key genes. These genes are closely linked to cancer metastasis, proliferation, apoptosis, cell cycle regulation, signal transduction, cancer microenvironment, immunotherapy, and neurodevelopment. Overall, the hub genes discovered in our study are pivotal in colon cancer and are anticipated to serve as important biological markers for the diagnosis and treatment of the disease.

## 1. Introduction

Colon cancer (CC) ranks as the fourth deadliest cancer globally [1]. The accumulation of diverse genetic and epigenetic alterations in colonic epithelial cells is a fundamental process underlying the onset and advancement of CC [2]. Colon cancer (CC) is a multifaceted and progressive condition that implicates numerous genes and stages. Several biomarkers associated with the survival and prognosis of CC have been investigated previously. Nonetheless, individual genes or biomarkers alone are insufficient for accurately predicting the outcomes of cancer patients [3,4].

The study involved analyzing the transcriptome sequencing data from TCGA-COAD and normal samples. Various methods were employed to analyze differentially expressed genes in CC, such as Gene Ontology (GO) and Kyoto Encyclopedia of Genes and Genomes (KEGG) pathway analysis, survival analysis, protein–protein interaction (PPI) network construction, LASSO regression analysis, tumor microenvironment analysis, Connectivity Map (cMAP) analysis, and ferroptosis analysis. These identified factors and pathways can function as biomarkers for cancer development and potential targets for the clinical treatment of CC.

## 2. Results

### 2.1. DEGs Identification

In this study, we conducted principal component analysis (PCA) on the TCGA-COAD dataset to differentiate between cancerous and normal tissues. The PCA plot displayed a clear separation between the two groups, suggesting distinct gene expression profiles in cancer and normal tissues (Figure 1B). After preprocessing the raw dataset, we utilized the three main R packages (DESeq2, edgeR, and limma) to identify differentially expressed genes (DEGs) in the TCGA-COAD dataset independently and create volcano plots (Figure 1A). The overlap of the DEGs identified by the three primary R packages was determined. Next, Venn diagrams were created separately for the upregulated and downregulated genes (Figure 1B). A heatmap was generated using the TCGA-COAD dataset to compare gene expression profiles between cancerous and normal tissues. The heatmap revealed distinct patterns of gene expression, emphasizing the discrepancies between cancer and normal tissues (Figure 1C).

### 2.2. Gene Set Enrichment Analysis

We performed GO enrichment and KEGG pathway analysis using the clusterProfiler package in R version is R 4.3.2 with 746 upregulated DEGs and 1083 downregulated DEGs. We visualizes the results using the ggplot2 package (Figure 2). Regarding biological process (BP) enrichment, the results suggest that DEGs are primarily involved in the production of molecular mediators of the immune response and immunoglobulin production (Figure 2A). For cellular component (CC) enrichment, DEGs were predominantly enriched in the immunoglobulin complex, collagen-containing extracellular matrix, apical plasma membrane, cell projection membrane, and monoatomic ion channel complex (Figure 2B). Regarding molecular function (MF) enrichment, DEGs were predominantly enriched in metal ion transmembrane transporter activity, monoatomic ion channel activity, glycosaminoglycan binding, extracellular matrix structural constituent, serine-type peptidase activity, serine hydrolase activity, serine-type endopeptidase activity, and metallopeptidase activity (Figure 2C). In the KEGG pathway analysis, the DEGs were grouped in the neuroactive ligand–receptor interaction, cytokine–cytokine receptor interaction, cAMP signaling pathway, calcium signaling pathway, and cell adhesion molecules (Figure 2D).

### 2.3. Survival Curves of 47 Hub Genes and Their Expression Levels in COAD

The survival curves and differential expression profiles of the 47 hub genes are presented in the Appendix A. Here, only the Kaplan–Meier survival curves and differential expression profiles of the important genes CDKN2A, CXCL1, CLCA1, MMP3, and MMP1 are shown (Figure 3). According to the research results, we observed that in CC tissues, the expression level of the CDKN2A gene exceeds the expression in normal tissues. However, the patients’ survival rates with low expression outperform those of patients with high expression. Likewise, the expression of the CXCL1 gene in CC exceeds that in normal tissues, and the patients’ survival rates with high expression are markedly higher than those with low expression. On the other hand, the expression level of the CLCA1 gene in CC tissues is lower than in normal tissues, but the patients’ survival rates with high expression are markedly higher than those with low expression. Additionally, the expression of the MMP3 and MMP1 genes is upregulated in CC tissues, and the patients’ survival rates with high expression are markedly higher than those with low expression.

### 2.4. PPI Network Construction

To explore the interactions among the 47 hub genes, we initially analyzed them using the STRING database and subsequently visualized their interactions as a protein–protein interaction (PPI) network in Cytoscape. The network comprises 29 nodes and 44 edges, as depicted in Figure 4A. We utilized cytoHubba to analyze the top 15 central genes in the PPI network (Figure 4B). Based on the findings from the protein–protein interaction network (PPI) analysis, we have identified notable interactions among 15 genes (CDKN2A, CXCL1, CLCA1, MMP3, MMP1, ITLN1, NME1, HEPACAM2, GRIK5, AOC3, ULBP2, ACSL6, STC2, GABRD, and CA4) in CC. These findings suggest that these genes may perform critical functions in the pathophysiological mechanisms of CC.

### 2.5. LASSO Regression Analysis

Following LASSO regression analysis, lasso.min identified 23 genes, whereas lasso.1se identified 18 genes (Figure 5A,B). We generated distinct AUC curves for lasso.min and lasso.1se (Figure 5C). We then plotted the AUC curves for the 23 selected genes by lasso.min at one, three, and five years (Figure 5D). We conducted a Kaplan–Meier survival analysis comparing high-risk and low-risk groups (Figure 5E). We generated a series of three linked risk factor plots, each displaying unique information: (a) predicted values for each patient sorted in ascending order, (b) patient survival time with color coding for living and deceased patients, and (c) heatmap illustrating the gene expression levels for selected genes in each sample (Figure 5F). Finally, we created a risk forest plot for the 23 selected genes (Figure 5G). After conducting LASSO regression, the AUC survival curve for the 23 selected genes was determined to be 0.81. The AUC prediction for multi-year survival rates yielded values of 0.8 at one year, 0.76 at three years, and 0.81 at five years. Upon stratifying the samples into high- and low-risk groups, it became evident from the Kaplan–Meier survival curves that the low-risk group exhibited notably higher survival rates than the high-risk group. Furthermore, a three-way interactive visualization of risk factors was developed. The reliability of the predictive model established by these 23 genes was assessed using a random forest model, revealing a very small *p*-value and a concordance index value of 0.79. These findings collectively indicate the high reliability of the predictive model.

### 2.6. Immune Cell Infiltration Analysis

To explore immune infiltration in cancerous and normal tissues within TCGA-COAD, we analyzed the processed expression matrix using the XCELL algorithm. We then created differential boxplots for 64 immune cell types (Figure 6A). We further investigated the relationship between the genes selected by lasso.min and the 64 immune cell types, generating a correlation heatmap (Figure 6B). Finally, we analyzed the differential expression of immune checkpoint genes in cancerous and normal tissues of TCGA-COAD and visualized the results with a corresponding lollipop chart (logFC ≥ 1, *p* < 0.05) (Figure 6C). According to the research results, we observed a significant decrease in CD8+ and CD4+ T cells in the adaptive immune cell population, while regulatory T cells (Tregs) significantly increased. In the myeloid immune cell population, M1 macrophages showed a significant decrease, whereas M2 macrophages exhibited a significant increase in CC. Furthermore, neutrophils, monocytes, dendritic cells, and mast cells showed a significant decrease, while natural killer (NK) cells showed an increase. In the analysis of the gene correlations selected by the LASSO model among 64 immune cells, we found that CCBE1, ZBTB7C, TPSG1, and CLDN23 were positively correlated with the majority of immune cells, whereas GABRE and TSPEAR were negatively correlated. In the analysis of immune checkpoint expression, we observed significant upregulation of TNFSF9, VTCN1, CD74, TDO2, TNFSF4, BTNL9, and CTLA4 in CC cells, while BTNL3, CEACAM1, CD209, CD160, KIR2DL4, BTLA, CD27, CD96, KIR3DL2, and CD40LG showed significant downregulation.

### 2.7. Mutation Analysis

To examine the gene mutation status in TCGA-COAD, we used the TCGAmutations package to retrieve and analyze the data, followed by generating an overview of the gene mutation landscape (Figure 7A). Additionally, we explored the mutation status of genes selected by lasso.min and depicted their respective mutation spectrum plot (Figure 7B). In the analysis of mutations, we found that missense mutations are the most prevalent in colorectal cancer, with SNP being the most common mutation type. The primary mutated genes comprise TTN, APC, MUC16, SYNE1, TP53, FAT4, and KRAS. Furthermore, ZBTB7C, WDR78, CCBE1, WDR72, and MMP3 demonstrate the highest mutation frequencies among the genes identified by the LASSO regression model.

### 2.8. Connectivity Map (cMAP) Analysis

To explore potential small molecule drugs with therapeutic potential in colon cancer patients, 150 upregulated and 150 downregulated differentially expressed genes (DEGs) were separately entered into the Connectivity Map (cMAP) database to identify small molecule compounds capable of reversing the expression changes linked to colorectal cancer-related pathogenic genes. After thorough analysis, the top nine compounds with the most negative scores, such as ISOX, vorinostat, NVP-AUY922, selumetinib, AS-703026, THM-I-94, NVP-TAE684, trichostatin-a, and scriptaid, were recognized as potential therapeutic agents for colon cancer treatment (Figure 8A). The chemical structures of these nine small molecule drugs are depicted in Figure 8B.

### 2.9. Ferroptosis Analysis

To explore the association between colon cancer and ferroptosis, we utilized FerrDB to gather ferroptosis-related genes. The expression variances of ferroptosis driver genes (Figure 9A) and suppressor genes (Figure 9B) in the TCGA-COAD dataset were visualized (logFC ≥ 1, *p* < 0.05). Subsequently, the associated protein–protein interaction (PPI) network was depicted (logFC ≥ 2, *p* < 0.05) (Figure 9C). CytoHubba was used to analyze the top 10 central genes in the PPI network (Figure 9D). In the ferroptosis-related analysis results, we found that in the ferroptosis-driving genes, the expression of genes such as H19, MIOX, ALOXE3, NOX4, PVT1, and CDKN2A was significantly upregulated, whereas LIFR, CPEB1, and MT1DP were significantly downregulated. In the ferroptosis-suppressing genes, genes such as CA9, ETV4, LINC01833, HCAR1, TFAP2A, and GDF15 were significantly upregulated, while MT1G and PDK4 genes were significantly downregulated. Additionally, in the context of the interaction of ferroptosis genes (logFC > 2) in CC, the top 10 most important genes were identified as CDKN2A, GDF15, MYCN, SCD, SLC7AL1, PDK4, NOX4, LCN2, CP, and CA9.

## 3. Discussion

In this study, we obtained and analyzed the TCGA-COAD dataset. Utilizing the differential analysis R packages DESeq2, edgeR, and limma, we identified 746 upregulated genes and 1083 downregulated genes, which constitute a highly reliable set of differentially expressed genes. Subsequently, we used these genes to address several questions through bioinformatics analyses: What are the significant Gene Ontology (GO) terms and KEGG signaling pathways in colon cancer (CC)? Which genes in these DEGs are linked to survival and act as hub genes? How do these hub genes interact? How can they be used for cancer prediction? What are the correlations between key genes from the prediction model and tumor microenvironment and gene mutations? How can drug screening be conducted based on these DEGs? Lastly, what is the relationship between CC and ferroptosis?

The analysis of biological process (BP) enrichment revealed that the differentially expressed genes (DEGs) are mainly linked to the production of molecular mediators involved in the immune response and immunoglobulin synthesis. These processes are closely associated with immune evasion in cancer. Producing immune response molecules is a crucial defense mechanism of the body against pathogens and cancer cells. Nevertheless, cancer cells can manipulate the production of immune response molecules and the generation of immunoglobulins through various mechanisms, allowing them to evade immune system attacks.

Within the cellular component (CC) enrichment analysis, the differentially expressed genes (DEGs) showed notable enrichment in the immunoglobulin complex, collagen-containing extracellular matrix, apical plasma membrane, cell projection membrane, and monoatomic ion channel complex. Immunoglobulin, a crucial blood protein, plays a significant role in the immune system by recognizing and eliminating pathogens and abnormal cells. Collagen, a vital structural protein in the extracellular matrix, is crucial for cell support and structure, and its abnormal accumulation may contribute to tumor invasion and metastasis. The apical membrane, a cellular surface membrane, is responsible for maintaining cell morphology and signal transduction, with its abnormal features potentially linked to enhanced invasiveness in specific cancer cells. The cell projection membrane, a cell surface structure, influences cell adhesion and movement, playing a crucial role in tumor invasion and metastasis. Ion channels, protein channels located on the cell membrane, control the ion balance within and outside the cell, with sodium channels playing a crucial role in tumor development.

In terms of molecular function (MF) enrichment, the differentially expressed genes (DEGs) exhibited significant enrichment in metal ion transmembrane transporter activity, monoatomic ion channel activity, glycosaminoglycan binding, extracellular matrix structural constituent, serine-type peptidase activity, serine hydrolase activity, serine-type endopeptidase activity, and metallopeptidase activity. Metal ions have a critical regulatory function within cells, and dysregulated expression of specific metal ion channel proteins may be linked to cancer development and progression through the modulation of intracellular and extracellular ion balance, impacting processes like cell proliferation and apoptosis. Monoatomic ion channel activity entails the specific transport of individual atomic ions (e.g., sodium, potassium, calcium) through channel proteins on the cell membrane. Aberrant ion channel activity can disturb intracellular and extracellular ion balance, potentially fostering cancer progression. Glycosaminoglycans are integral components of the extracellular matrix, and aberrant glycosaminoglycan binding capacity may be associated with tumor invasion and metastasis. Certain glycosaminoglycan receptors on cancer cell surfaces can enhance tumor cell adhesion, invasion, and metastasis. The extracellular matrix provides structural support outside cells, consisting of various proteins and polysaccharides. Defects in extracellular matrix components may promote tumor invasion and metastasis by influencing cell–tissue interactions. With regard to serine protease activity, serine hydrolase activity, serine endopeptidase activity, and metallopeptidase activity, these enzymes participate in protein degradation and modification. Dysregulated protease activities can disrupt intracellular signaling pathways, promoting cancer progression.

The KEGG pathway analysis indicated a significant association of the DEGs with neuroactive ligand–receptor interaction, cytokine–cytokine receptor interaction, the cAMP signaling pathway, the calcium signaling pathway, and cell adhesion molecules. The neuroactive ligand-receptor interaction and cytokine-cytokine receptor interaction pathways are vital in cell signaling, governing growth factors, neurotransmitters, and hormones. Dysregulated signaling may result in uncontrolled cell proliferation, inhibited apoptosis, and tumor formation. The cAMP signaling pathway and calcium signaling pathway regulate intracellular signal transduction and modulate intracellular calcium ion levels. Abnormal cAMP and calcium signaling are closely associated with cancer initiation and progression. Cell adhesion molecules are crucial for cell–cell adhesion and interaction, influencing processes such as cell migration, invasion, and metastasis. These processes play a crucial role in tumor development and dissemination.

After conducting the log-rank test and single-factor Cox regression analysis (with log rank *p* < 0.05 and Cox results *p* < 0.05), a total of 47 survival-related genes were identified from the highly credible DEGs. By analyzing the protein–protein interaction (PPI) network, 15 central genes were identified among the 47 hub genes.

The onset of CC involves numerous factors and genes. Processes such as cell growth, apoptosis, DNA repair, and signal transduction can be impacted by abnormal gene expression and mutations, resulting in different molecular mechanisms that influence the onset of CC. The CDKN2A gene encodes two tumor suppressor proteins, p16 and p14, which play crucial roles in regulating the cell cycle and metabolism in melanoma. Furthermore, these proteins are closely linked to immune infiltration in the tumor microenvironment [5]. Elevated CDKN2A expression in CC is linked to an unfavorable prognosis. CXCL1 can promote the migration and invasiveness of breast cancer by activating the transcription of SOX4 via the NF-κB pathway, resulting in the subsequent epithelial-mesenchymal transition (EMT) process [6]. CXCL1 expression is increased in CC. Elevated CLCA1 expression levels can inhibit the invasiveness of colorectal cancer (CRC). CLCA1 may function as a tumor suppressor by inhibiting the Wnt/β-catenin signaling pathway and the epithelial-mesenchymal transition (EMT) process [7]. Reduced CLCA1 expression in CC is linked to a favorable prognosis. MMP1 plays a crucial role in promoting tumor progression in large cell carcinoma of the lung by inducing fibroblast senescence [8]. MMP3 initially collaborates with oncogenic KRAS to drive tumorigenesis in pancreatic cancer and activate the stromal microenvironment. Subsequently, it becomes a key driver in promoting tumor invasion and progression [9]. Ovarian cancer (OC) cells have been observed to modify mesothelial cells in visceral adipose tissue by downregulating ITLN1, thereby enhancing the invasion potential and proliferation of OC cells in the omental microenvironment [10]. These genes collectively influence the onset of CC through their involvement in the cell cycle, inflammatory response, intercellular signaling, and the tumor microenvironment. Their abnormal expression and regulatory relationships constitute the complex mechanisms underlying the occurrence, development, and metastasis of CC. In-depth exploration of these mechanisms is essential for understanding the onset process of CC, identifying potential therapeutic targets, and developing personalized treatment strategies.

In the field of CC research, accurately predicting the survival prognosis of patients is crucial for developing personalized treatment plans and providing appropriate medical care. The objective of this study is to utilize the LASSO regression model to establish a multi-gene prediction model for predicting the survival prognosis of CC patients. Through the analysis and integration of multiple genes related to the survival prognosis of CC patients, we aim to provide a reliable prediction tool for clinical practice, in order to better understand the disease progression of patients and support medical decision-making, with the goal of early detection leading to early treatment. In this study, we used the LASSO regression model to select and evaluate CC-related genes to identify those with significant predictive capabilities for patient survival prognosis. Compared to traditional prediction models, the LASSO regression model is highly favored for its ability to effectively handle high-dimensional data and reduce model complexity. Through this model, we successfully identified a set of key genes closely associated with the survival prognosis of CC patients, providing important gene markers for the construction of the prediction model. Furthermore, our research revealed that, compared to previous studies, our analysis not only clarified the key genes related to the survival prognosis of CC but also uncovered the underlying potential mechanisms behind CC survival prognosis, providing valuable clues for further in-depth research. Our findings not only expanded the understanding of CC survival prognosis but also offered a new perspective for personalized treatment of CC in terms of potential drug targets or diagnostic markers.

The tumor microenvironment (TME) is the origin and residence of tumor cells, comprising not only the tumor cells themselves but also neighboring cells like fibroblasts, immune cells, inflammatory cells, and vascular cells collectively known as cancer-associated stromal and immune cells. Moreover, it includes the secretory products of these cells, such as cytokines, chemokines, and non-cellular components of the extracellular matrix (ECM). Tumor growth and metastasis are intricately linked to the surrounding environment. Key features of the tumor microenvironment include hypoxia, chronic inflammation, and immune suppression, which collaboratively enhance the development and growth of tumor cells. During tumor development, local immune cells play a pivotal role in shaping the composition of the tumor microenvironment. Different cell types in the TME can display either tumor-suppressive or tumor-supportive properties. The diverse immune and stromal cells in the TME, along with their secretory products and the extracellular matrix, collectively impact tumor development [11].

There is a notable decrease in CD8+ and CD4+ T cells, along with a significant increase in regulatory T cells (Tregs), in the adaptive immune cell population in CC. In the myeloid immune cell population, M1 macrophages show significant downregulation, whereas M2 macrophages exhibit significant upregulation in CC. Moreover, neutrophils, monocytes, dendritic cells, and mast cells demonstrate significant downregulation, while natural killer (NK) cells show upregulation in CC across both adaptive and innate immune cell populations (Figure 6A). These findings illuminate the immune cell composition in the tumor microenvironment and its potential influence on tumor progression. The alterations in the immune cell composition documented in CC suggest a complex interplay between the tumor and the immune system, which may contribute to the immunosuppressive characteristics of the tumor microenvironment in CC. Moreover, these insights may have implications for the development of targeted immunotherapies to modulate the immune landscape in CC. We can target the disparities in immune cell infiltration between colon cancer and normal intestinal tissue by developing targeted drugs that enhance the activity of specific immune cells, such as NK cells, CD8 T cells, etc., amplifying the activity and quantity of these immune cells and consequently eradicating cancer cells to achieve effective cancer treatment. Immune checkpoint genes (ICGs) are vital in evading self-reaction and serve as novel targets for developing cancer treatment methods [12]. We can observe significant up-regulation of TNFSF9 and VTCN1, while BTNL3, CEACAM1, CD209, CD160, and KIR2DL4 are significantly down-regulated (Figure 6C). TNFSF9 is significantly up-regulated in pancreatic cancer (PC) and may promote the growth and metastasis of PC in vivo and in vitro through the Wnt/Snail signaling pathway. Additionally, TNFSF9 can induce the M2 polarization of macrophages by activating the Wnt signaling in pancreatic cancer cells, thereby promoting the metastasis of PC [13]. VTCN1 (B7-H4) is highly expressed in many tumor tissues. The biological activity of B7-H4 is associated with a reduced inflammatory CD4 T cell response, and the correlation between tumor-associated macrophages expressing B7-H4 and regulatory T cells (Tregs) expressing FoxP3 in the tumor microenvironment [14]. The human intestinal epithelial cells express BTNL3, which induces a selective TCR-dependent response in human colonic Vγ4 cells [15]. CEACAM1 serving as an allosteric ligand of TIM-3, is essential for its ability to mediate T cell suppression. This interaction plays a crucial role in regulating both self-immunity and anti-tumor immunity [16]. In small cell lung cancer, M1 macrophages are up-regulated in the CD209-High group. The activation of the CD209 signaling pathway is associated with increased infiltration of CD8 T cells, and the activation of the CD209 signaling pathway is also associated with increased neutrophil infiltration [17]. In patients with hepatocellular carcinoma (HCC), the intra-tumoral expression of CD160 is decreased in NK cells, but not in CD8+ T cells [18]. Knocking down KIR2DL4 in human NK cells in vitro can inhibit their cytotoxicity and also suppress the secretion of tumor necrosis factor α and interferon γ. Conversely, upregulation of KIR2DL4 can activate the MEK/ERK signaling pathway, which constitutes an activation pathway for NK cells [19]. Human intestinal epithelial cells express BTNL3, which triggers a selective TCR-dependent response in human colonic Vγ4 cells. CEACAM1 acts as an allosteric ligand of TIM-3, essential for mediating T cell suppression. Conversely, upregulation of KIR2DL4 can activate the MEK/ERK signaling pathway, an activation pathway for NK cells. Immune checkpoint (ICG) therapy is an emerging cancer treatment method that modulates the immune system to suppress tumor growth. This form of treatment involves the use of medications to block tumor cells from evading immune recognition, enabling the immune system to identify and combat the tumor cells. The approach has been widely implemented across various cancer types and has demonstrated promising therapeutic effects. The underlying principle of immune checkpoint therapy is that the immune system is capable of recognizing and eliminating abnormal cells, including tumor cells, under normal circumstances. However, tumor cells often employ immune checkpoints to evade immune system attacks. Immune checkpoints represent a molecular signaling system that regulates immune system activity, preventing it from targeting normal tissues. Through interaction with these checkpoints on immune cells, tumor cells can evade immune recognition and subsequent attack. The key to immune checkpoint therapy lies in blocking these signals, thus reversing immune system suppression and reinstating its ability to target and attack tumor cells. An important advantage of immune checkpoint therapy is its lasting therapeutic effects. In comparison to conventional treatments such as radiotherapy and chemotherapy, this approach not only diminishes tumor volume but also triggers sustained immune responses against the tumor. This strategy can lead to the development of targeted therapeutic drugs for alterations in immune checkpoint expression in colon cancer, offering newfound hope and opportunities for numerous CC patients, alleviating their suffering.

Effective drug therapies for CC treatment are currently insufficient. Thus, there is an urgent need to investigate potential drugs for this purpose. Our study offers a fresh perspective by utilizing cMAP analysis to link CC-related pathogenic genes in the search for potential compounds for CC treatment. Through cMAP analysis, we have identified candidate drugs, including ISOX, vorinostat, NVP-AUY922, selumetinib, AS-703026, THM-I-94, NVP-TAE684, trichostatin-a, and scriptaid. It is worth noting that ISOX exhibits the highest negative enrichment scores in the cMap analysis, indicating its strong potential in reversing the expression of relevant pathogenic genes in CC.

Notably, in cMap analysis, ISOX shows the highest negative enrichment scores, suggesting a significant reversal of the expression of pathogenic genes in CC. ISOX, also known as CAY10603, is a selective inhibitor of histone deacetylase 6 (HDAC6) [20]. ISOX significantly inhibits the survival of osteosarcoma cells in a dose-dependent manner. It also dose-dependently inhibits proliferation, colony formation, migration, and invasion. Further in vivo experiments using animal models demonstrate that ISOX treatment significantly suppresses tumor growth. Flow cytometry analysis indicates that ISOX treatment induces increased infiltration of CD8+ T cells into the tumor [21]. ISOX inhibits HDAC6, leading to a significant reduction in c-Jun N-terminal kinase (JNK) and c-Jun phosphorylation, preceding its inhibitory effect on the growth of glioma cells. These effects are attributed to the HDAC6 inhibitor-induced inhibition of mitogen-activated protein kinase 7 (MKK7), which has been identified as crucial in JNK activation and carcinogenesis in glioma cells [22]. Thus, it is speculated that early administration of ISOX in CC patients may inhibit the onset and advancement of the disease, consequently leading to a substantial extension of patients’ lifespans.

Ferroptosis, an iron-dependent regulated cell death pathway induced by the toxic buildup of lipid peroxides on cell membranes, shows significant promise in cancer treatment. Our study revealed the critical involvement of CDKN2A in CC. Elevated CDKN2A expression in CC is notably linked to a poor prognosis (Figure 3). Furthermore, we observed a close association between CDKN2A and ferroptosis (Figure 9D). The CDKN2A gene encodes two proteins, p14 and p16. p14 determines cell fate by indirectly stabilizing p53, while p16 suppresses tumor formation by inhibiting CDK4/6 [23,24]. The expression of the CDKN2A gene can result in cell cycle arrest at the G1 phase, leading to the inhibition of cell proliferation and the promotion of tumor cell apoptosis [25]. Research has indicated that the loss of the CDKN2A gene alters the lipid composition of glioblastoma multiforme (GBM), rendering GBM cells sensitive to lipid peroxidation and ferroptosis. This loss also reduces the storage of oxidative polyunsaturated fatty acids (PUFAs) in lipid droplets. Furthermore, the loss of P16 alone is sufficient to make GBM cells sensitive to ferroptosis [26]. We can consider developing targeted drugs against the CDKN2A gene, which could promote the development of cancer cells towards the ferroptosis pathway. This novel treatment approach has the potential to lead the way in revolutionizing the field of cancer treatment, bringing new hope and possibilities for patients. This cutting-edge research will not only drive innovation in treatment methods but also pave the way for the application of ferroptosis in cancer treatment, offering patients more possibilities and hope.

## 4. Materials and Methods

### 4.1. Data Collection

The mRNA expression profiles of 471 Colon Adenocarcinoma (COAD) and 41 normal tissues were obtained from the TCGA Genomic Data Commons (GDC) database (https://portal.gdc.cancer.gov/, accessed on 17 December 2023). To retrieve and organize data, the R package TCGAbiolinks can be employed. However, detailed clinical and pathological information was available for only 424 colon cancer samples, and the comprehensive clinical characteristics of these patients are presented in Table 1. Patients with missing or incomplete follow-up data were excluded from the survival analysis. Subsequently, survival analysis for 424 COAD patients was further performed to investigate the relevant differentially expressed genes.

### 4.2. Performing Differential Gene Expression Analysis Using the Three Major R Packages, DESeq2, edgeR, and Limma

DESeq2, edgeR, and limma are different R packages for downstream differential gene expression analysis. DESeq2 (differential expression analysis of RNA-Seq data) is based on a negative binomial distribution model. It considers differences between samples and the variability of gene expression using the Bayesian method. edgeR (Empirical Analysis of Digital Gene Expression Data in R) is also based on a negative binomial distribution model. It uses a Bayesian method to improve stability of estimates by adapting the estimation of within-group variability. Limma (linear models for microarray analysis) is based on a linear model and uses the Bayesian method to estimate differential variances for each gene. Utilizing three distinct R packages for the screening of differentially expressed genes is intended to improve the credibility and robustness of the findings, thereby facilitating more accurate subsequent analyses, including enrichment and survival analyses, thus yielding more precise results. Genes exhibiting a Log2-fold change ≥ 2 were categorized as differentially expressed genes (DEGs), with statistical significance in gene expression determined by a *p* value < 0.05. We classified the differential analysis results from the three major R packages into upregulated and downregulated genes and took the intersection. The genes from this intersection were used for subsequent analyses.

### 4.3. Functional Enrichment Analysis

We performed Gene Ontology (GO) and Kyoto Encyclopedia of Genes and Genomes (KEGG) pathway enrichment analyses to ascertain the biological functions of the differentially expressed genes (DEGs). Visualizations were conducted using R packages such as clusterProfiler and ggplot2, with a significance threshold set at *p* < 0.05, FDR < 0.01 (Table 2).

### 4.4. Survival Analysis

We performed survival analysis using two different methods, the log-rank test and single-factor Cox regression analysis, to select survival-related genes (log_rank_*p* < 0.05 and cox_results_*p* < 0.05). We conducted KM analysis and drew survival curves using the R packages survival and survminer. We then took the intersection of the selected survival-related genes and the previously identified DEGs to obtain 47 hub genes. We utilized the ggplot2 package to illustrate the KM plots of these 47 hub genes and visualized the expression levels of these genes in the TCGA-COAD dataset. The enrichment status of each entry in the GO analysis of these 47 hub genes has been summarized (Table 3).

### 4.5. PPI Network Construction

After inputting 47 hub genes into STRING (https://string-db.org/, accessed on 30 January 2024) (version 12.0), a protein–protein interaction (PPI) network was predicted. The minimum required interaction score was based on 0.15. Then, Cytoscape (version 3.10.0) was used to visualize the PPI network. We adjusted the node sizes in the PPI network based on their degrees, using log2FC to change node colors (red for upregulation and blue for downregulation), and employed the combined_score to regulate the thickness and color gradient of the edges. To identify crucial nodes in a biological network, the cytoHubba plugin can implement the degree algorithm to assess the connectivity of network nodes. This method aids in the identification of highly connected nodes that often possess significant influence within the network. Leveraging cytoHubba and the degree algorithm, we can gain vital insights into the structural and functional attributes of biological networks, thereby contributing to the discovery of potential biomarkers, drug targets, and disease-associated genes.

### 4.6. LASSO Regression Analysis

We performed LASSO (Least Absolute Shrinkage and Selection Operator) regression analysis on the 47 hub genes. We used the glmnet package to build a LASSO model. We then chose an appropriate lambda value to build the model, where the size of lambda determines the number of genes selected for the model. Two dashed lines indicate two special lambda values: lambda.min and lambda.1se. The lambda values between these two are considered suitable. The model built with lambda.1se is the simplest, with fewer genes used, while lambda.min has slightly higher accuracy and uses a greater number of genes. We chose to use lambda.min to build the model. We used the pROC and ggplot2 packages to plot ROC curves for lambda.min and lambda.1se. The AUC value, ranging from 0 to 1, reflects the model’s performance—closer to 1 indicates better performance. We then utilized the survminer, survival, and timeROC packages to plot time-ROC curves. We employed the ggrisk package to construct a linked three-panel plot of risk factors.

### 4.7. Immune Cell Infiltration Analysis

We utilized the IOBR package to perform immune infiltration analysis and employed the xCell method to calculate immune cell infiltration, thereby exploring the immune microenvironment of the disease [27]. The results of immune infiltration were visualized using the ggplot2 package. We identified immune checkpoint genes through a literature review and studied their differential expression in TCGA-COAD. We then utilized ggplot2 and ggpubr for visualization.

### 4.8. Mutation Analysis

We used the R package TCGAmutations from GitHub to retrieve the data and visualized the data using the maftools package.

### 4.9. Connectivity Map (cMAP) Analysis

CMAP (https://clue.io) is a gene expression signature-based database that elucidates relationships between diseases, genes, and small molecule compounds [28,29,30,31]. In this study, 150 upregulated and 150 downregulated differentially expressed genes (DEGs) from the TCGA-COAD dataset were separately incorporated into the cMAP online database to identify potential small molecule drugs for the treatment of colon cancer. Ultimately, nine compounds with an enrichment score less than −90 were identified.

### 4.10. Ferroptosis Analysis

Ferroptosis is an iron-dependent regulated cell death mechanism closely associated with cancer. The process of ferroptosis is complex. In order to investigate the relationship between colon cancer and ferroptosis, we downloaded the driver genes and suppressor genes from the FerrDB (http://www.zhounan.org/ferrdb/current/) database for analysis of the expression changes in these genes. We then used ggplot2 and ggpubr packages for visualization.

## 5. Conclusions

In summary, we conducted differential gene expression (DEGs) and enrichment analyses utilizing the TCGA-COAD dataset. We developed a survival-related gene risk model through LASSO regression to accurately predict the prognosis of CC patients. Furthermore, we investigated the immune microenvironment, predicted small molecule drugs, and explored ferroptosis in CC. These results can significantly contribute to the development of therapeutic drugs for CC.

## Figures and Tables

**Figure 1 ijms-25-03954-f001:**
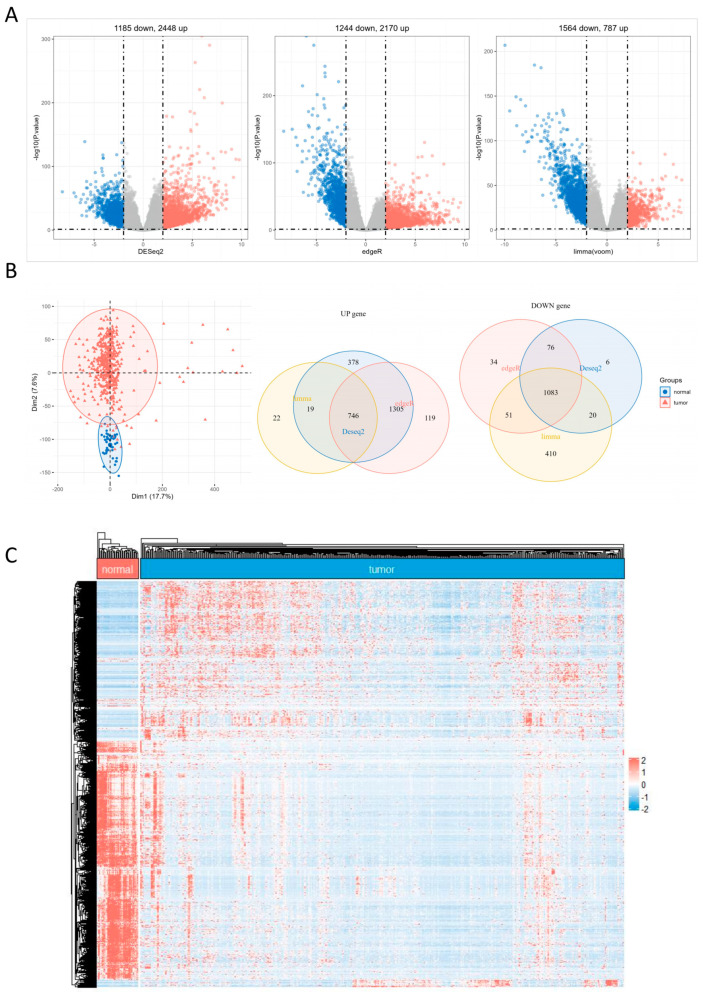
DEGs identification. (**A**) The three volcano plots were generated using the TCGA-COAD dataset by the three major R packages (DESeq2, edgeR, and limma). In the plots, red represents the significantly upregulated genes, and blue represents the significantly downregulated genes. (**B**) The PCA analysis plot of the TCGA-COAD dataset and the Venn diagram of differentially expressed genes (DEGs) from the three major R packages. (**C**) Heatmap of DEGs in the TCGA-COAD dataset.

**Figure 2 ijms-25-03954-f002:**
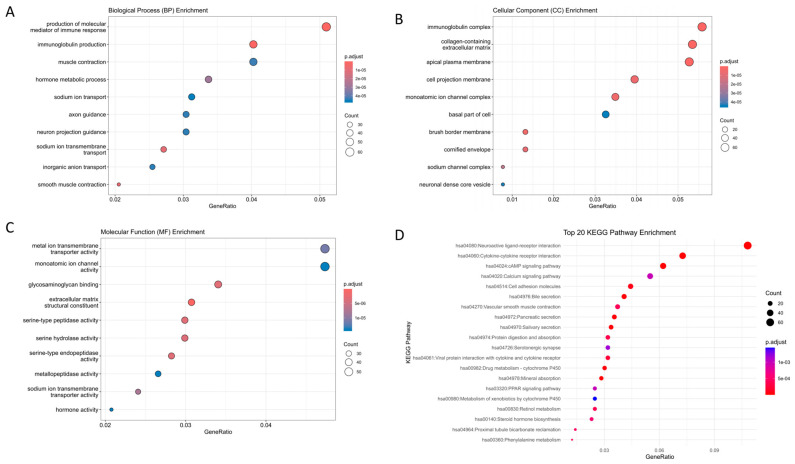
Gene set enrichment analysis. Performing an enriched analysis of common differentially expressed genes (DEGs) in terms of Gene Ontology (GO) for biological process (BP), cellular component (CC), and molecular function (MF), as well as Kyoto Encyclopedia of Genes and Genomes (KEGG) signaling pathways. (**A**) GO-BP. (**B**) GO-CC. (**C**) GO-MF. (**D**) KEGG.

**Figure 3 ijms-25-03954-f003:**
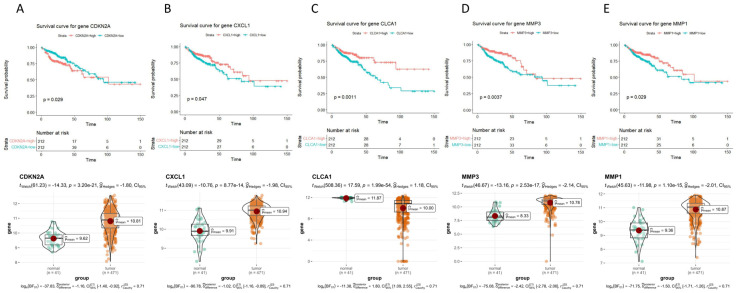
Survival curves and differential expression profiles of important genes. (**A**) CDKN2A. (**B**) CXCL1. (**C**) CLCA1. (**D**) MMP3. (**E**) MMP1.

**Figure 4 ijms-25-03954-f004:**
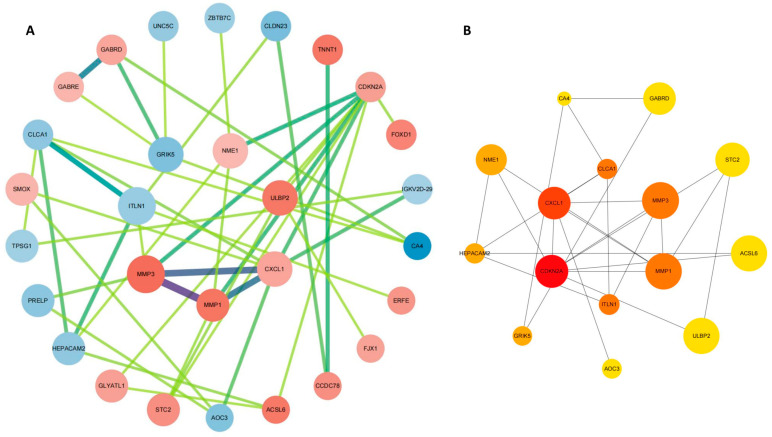
PPI network construction. (**A**) Protein–protein interaction (PPI) network of the 47 hub genes. (**B**) cytoHubba was performed to screen the top 15 genes based on previous PPI networks.

**Figure 5 ijms-25-03954-f005:**
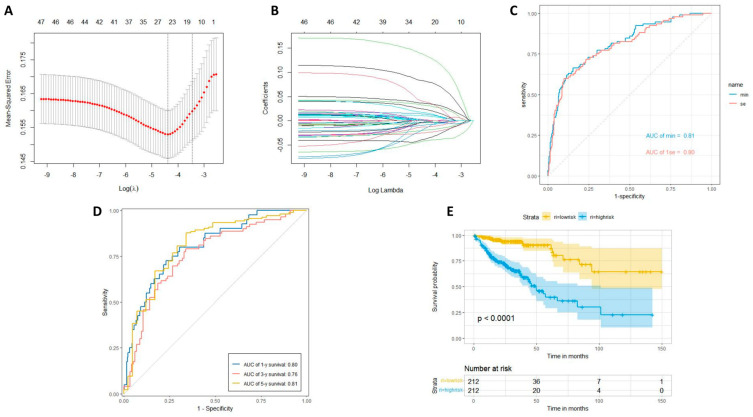
LASSO regression analysis. (**A**,**B**) LASSO regression analysis log lambda. (**C**) The AUC curves for lasso.min and lasso.1se. (**D**) The AUC curves for lasso.min chose genes at one year, three years, and five years. (**E**) Kaplan–Meier survival analysis between the high-risk and low-risk groups. (**F**) A linked set of three risk factor plots. (**G**) Cox-forest. (* *p* < 0.05, ** *p* < 0.01, *** *p* < 0.001).

**Figure 6 ijms-25-03954-f006:**
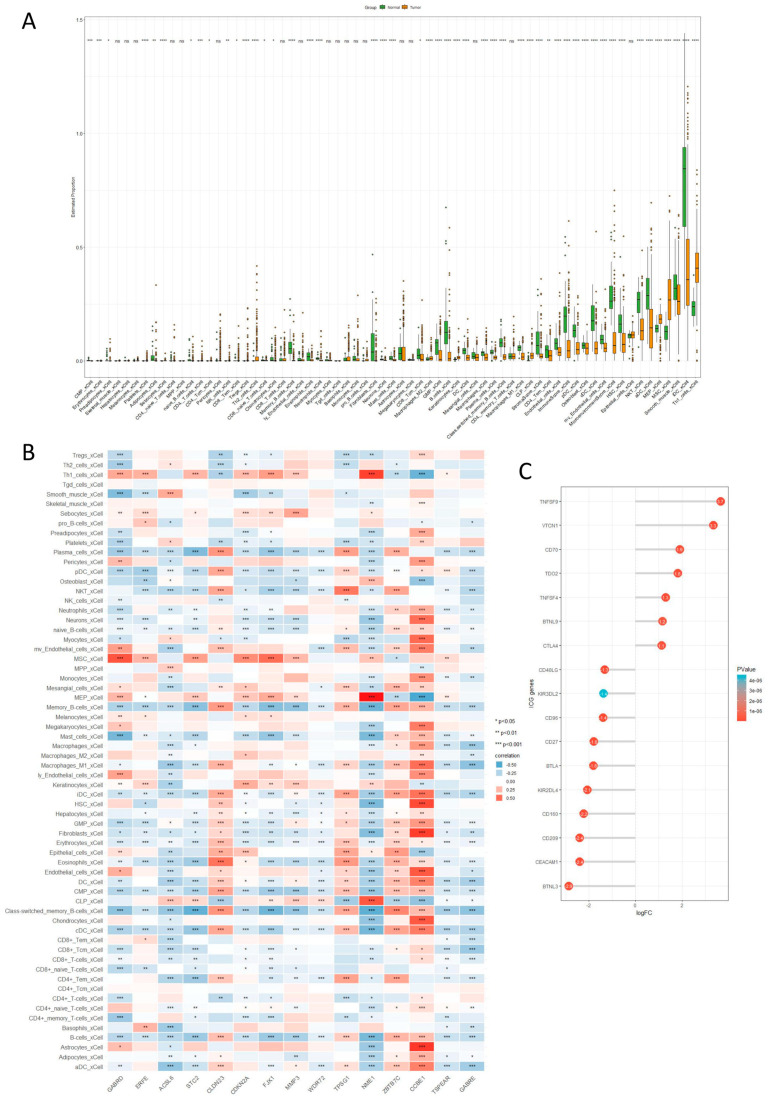
Immune cell infiltration analysis. (**A**) Differential boxplots of the 64 immune cell types assessed by the XCELL algorithm. (**B**) Correlation heatmap between genes selected by lasso.min and the 64 immune cell types. (**C**) Lollipop chart depicting the differential expression of immune checkpoint genes.(* *p* < 0.05, ** *p* < 0.01, *** *p* < 0.001, **** *p* < 0.0001).

**Figure 7 ijms-25-03954-f007:**
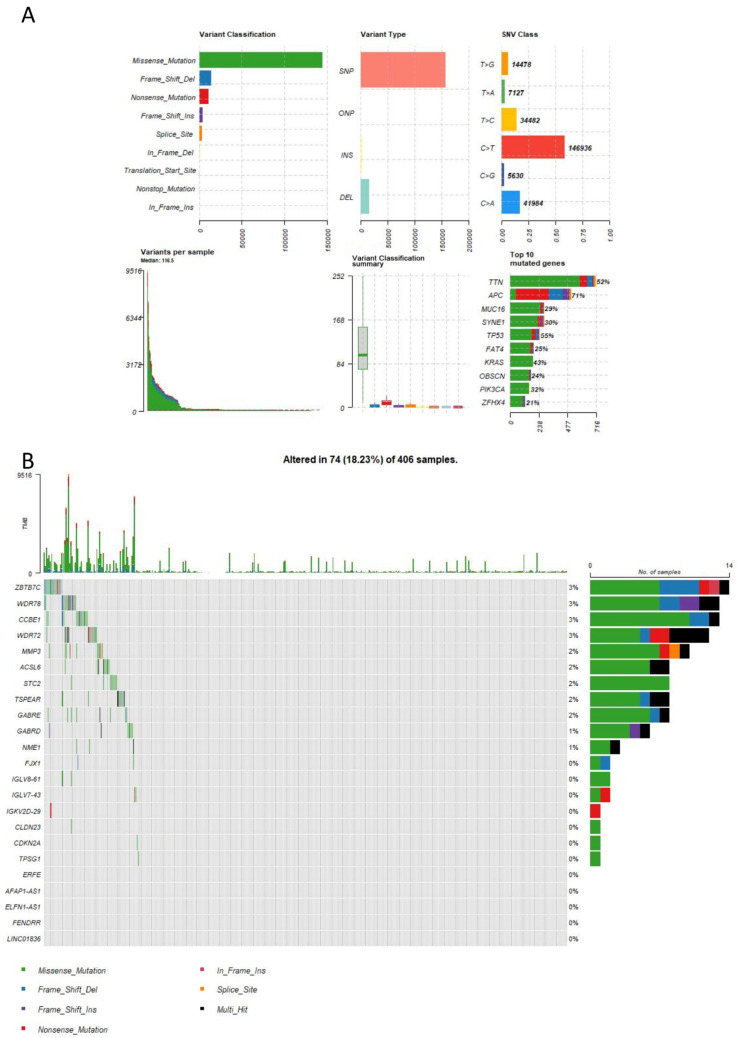
Mutation analysis. (**A**) Overview plot of gene mutations. (**B**) Mutation spectrum plot of genes selected by lasso.min.

**Figure 8 ijms-25-03954-f008:**
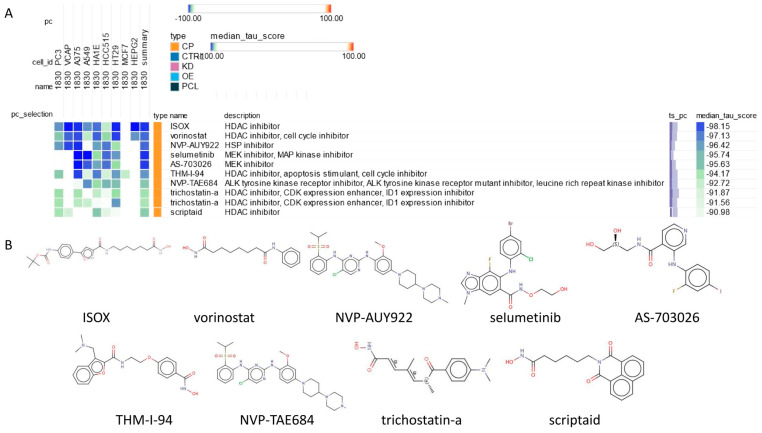
Connectivity map (cMAP) analysis. (**A**) Heatmap of significant perturbation compounds. (**B**) The chemical structures of those nine compounds.

**Figure 9 ijms-25-03954-f009:**
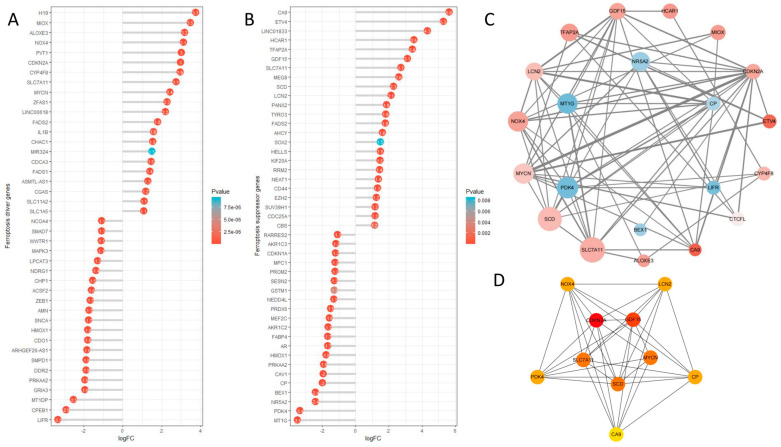
Ferroptosis analysis. (**A**) Lollipop chart depicting the differential expression of ferroptosis driver genes. (**B**) Lollipop chart depicting the differential expression of ferroptosis suppressor genes. (**C**) Ferroptosis corresponding protein–protein interaction (PPI) network. (**D**) Visualization of the PPI network for the top 10 genes associated with ferroptosis.

**Table 1 ijms-25-03954-t001:** Clinical features in COAD patients.

Variables		Patients	Percentages (%)
gender	male	298	52.19
	female	271	47.46
A/D	Alive	442	77.41
	Dead	127	22.24
Age, years	≦69	272	47.80
	>69	297	52.20
T stage	T1	11	1.93
	T2	95	16.64
	T3	390	68.30
	T4	39	6.83
N stage	N0	335	58.67
	N1	129	22.59
	N2	105	18.39
M stage	M0	416	72.85
	M1	81	14.19
	MX	64	11.21

**Table 2 ijms-25-03954-t002:** Important terms in GO and KEGG enrichment results.

	Name	Count	%	*p*-Value	Genes
GO-BP	production of molecular mediator of immune response	62	5.09	9.13 × 10^−13^	CD22, PGC, SLC7A5, AICDA, CD160, CCR2, TREM1, VPREB3, CD36, APOA2, SLAMF9, TLR3, KLK7, MZB1, KIR2DL4, ELANE, IGKV4-1, IGKV6-21, IGKV3D-20, IGKV3D-11, IGKV1D-42, IGLV4-69, IGLV8-61, IGLV4-60, IGLV10-54, IGLV1-50, IGLV5-48, IGLV7-46, IGLV5-45, IGLV1-44, IGLV7-43, IGLV2-33, IGLV2-14, IGLV3-10, IGLV3-9, IGLV4-3, TRDV1, IGKV3D-15, IGKV6D-21, IGKV2D-30, IGKV1-6, IGKV3-20, IGKV1D-33, IGKV1-17, IGKV1-8, IGKV1-16, MIF, IGKV2-24, IGKV2D-24, IGKV1-9, IGKV1-39, IGKV2D-28, IGKV1D-17, IGKV3-7, IGKV2-30, IGKV2D-29, IGKV1-12, IGKV2-28, IGKV1-27, IGKV1D-39, IGLV2-8, IGKV1D-12
	immunoglobulin production	49	4.03	1.29 × 10^−17^	CD22, AICDA, VPREB3, MZB1, IGKV4-1, IGKV6-21, IGKV3D-20, IGKV3D-11, IGKV1D-42, IGLV4-69, IGLV8-61, IGLV4-60, IGLV10-54, IGLV1-50, IGLV5-48, IGLV7-46, IGLV5-45, IGLV1-44, IGLV7-43, IGLV2-33, IGLV2-14, IGLV3-10, IGLV3-9, IGLV4-3, TRDV1, IGKV3D-15, IGKV6D-21, IGKV2D-30, IGKV1-6, IGKV3-20, IGKV1D-33, IGKV1-17, IGKV1-8, IGKV1-16, IGKV2-24, IGKV2D-24, IGKV1-9, IGKV1-39, IGKV2D-28, IGKV1D-17, IGKV3-7, IGKV2-30, IGKV2D-29, IGKV1-12, IGKV2-28, IGKV1-27, IGKV1D-39, IGLV2-8, IGKV1D-12
GO-CC	immunoglobulin complex	72	5.58	5.46696 × 10^−49^	CD79A, JCHAIN, IGKV4-1, IGKV6-21, IGKV3D-20, IGKV3D-11, IGKV1D-42, IGLV4-69, IGLV8-61, IGLV4-60, IGLV10-54, IGLV1-50, IGLV5-48, IGLV7-46, IGLV5-45, IGLV1-44, IGLV7-43, IGLV2-33, IGLV2-14, IGLV3-10, IGLV3-9, IGLV4-3, IGLC7, IGHA2, IGHA1, IGHV6-1, IGHV2-5, IGHV3-7, IGHV3-11, IGHV3-13, IGHV3-15, IGHV3-21, IGHV3-23, IGHV3-35, IGHV4-39, IGHV3-48, IGHV3-49, IGHV5-51, IGHV3-53, IGHV1-58, IGHV3-66, IGHV3-73, IGKV3D-15, IGHV4-59, IGHV3-74, IGKV6D-21, IGHV3-72, IGKV2D-30, IGKV1-6, IGKV3-20, IGKV1D-33, IGKV1-17, IGKV1-8, IGKV1-16, IGKV2-24, IGKV2D-24, IGKV1-9, IGKV1-39, IGKV2D-28, IGKV1D-17, IGKV3-7, IGKV2-30, IGKV2D-29, IGKV1-12, IGKV2-28, IGKV1-27, IGKV1D-39, IGLL5, IGLV2-8, IGKV1D-12, IGHV7-4-1, IGHV3-64D
	collagen-containing extracellular matrix	69	5.35	3.37778 × 10^−13^	COL11A1, NTN1, FGFR2, COL19A1, ADAMTS2, CMA1, CTSG, BMP7, LAMA1, SRPX, SRPX2, SFRP1, COMP, WNT2, PTPRZ1, SERPINE1, OGN, CXCL12, COL1A1, COL7A1, ANGPTL1, MFAP2, MMP8, TGFBI, CLU, ITIH5, COL10A1, F13A1, AMELX, FGL2, GDF15, MATN3, ADAMDEC1, CILP, MMRN1, INHBE, DPT, AHSG, HAPLN1, HMCN2, NCAM1, SPARCL1, ABI3BP, ACAN, AZGP1, CLEC3B, EDIL3, SHH, CTHRC1, VWA2, MFAP4, KRT1, TNXB, FGB, ANGPTL7, COL6A5, ZG16, F2, BGN, EMILIN3, ANGPTL5, VWC2, PRELP, ELANE, COL4A6, MFAP5, EGFL6, VIT, MMP28
	apical plasma membrane	68	5.28	9.77371 × 10^−14^	SLC13A2, CEACAM7, DPEP1, CLCA4, CASR, SLC9A3, CYBRD1, ABCB11, CDHR2, CA12, CEACAM1, PTPRH, KCNK2, SLC15A1, SLC4A11, SI, SLC26A3, CDHR5, SLC7A5, AQP8, FOLR1, CLIC5, SLC9A2, PAPPA2, ABCG2, TRPM6, ECRG4, CNTFR, SLC17A1, ATP1B2, SLC6A6, SLC14A2, CBLIF, CD36, PRKG2, SLC4A10, ANK2, SLC17A4, ATP6V0D2, SLC26A2, KCNMA1, KCNB1, SLC5A11, STC1, IL6R, PTH1R, CD300LG, AQP5, CLDN1, SCNN1G, MYO1A, CA4, NAALADL1, SLC22A11, SCNN1B, P2RY1, SLC23A1, KISS1, MAL, SPTBN2, SLC6A19, SLC26A9, OXTR, SAPCD2, P2RY4, P2RX2, SLC6A14, GPIHBP1
	cell projection membrane	51	3.95	6.88773 × 10^−8^	DPEP1, PHLPP2, PSD, SLC9A3, CNGB1, CYBRD1, CDHR2, ITGA8, FAP, CEACAM1, PTPRH, EPB41L3, TESC, SLC26A3, CDHR5, BMX, GABRE, SLC7A5, AQP8, CA9, FOLR1, BVES, GABRG2, ABCG2, TRPM6, ATP1B2, SLC6A6, PDE6A, CD36, SLC17A4, SLC7A11, SLC26A2, KCNB1, PDE9A, S100P, HHIP, GPER1, PRKCB, CA4, FAM107A, P2RY12, DRD5, GAP43, SLC6A19, TSPEAR, ITLN1, DDN, MAPT, CYS1, NME1, SSTR3
	monoatomic ion channel complex	45	3.49	4.70643 × 10^−8^	BEST2, TRPC7, CNGB1, KCNK2, GABRE, SLC17A7, GRIN2D, GRIK5, CLIC5, GABRG2, CASQ2, OLFM3, BEST3, DPP6, SCN7A, CACNG8, BEST4, CNGA3, KCNMB1, SCN2B, KCNA6, GRIA4, SCN3A, KCNJ16, KCNMA1, KCNB1, GRIK3, HTR3A, SCNN1G, SCN11A, SCNN1B, SCN9A, LRRC8E, KCNG3, SCN4B, KCNA3, SLC17A8, GRIN2A, CLCNKB, KCNIP4, HTR3E, GABRD, VWC2, TMEM249, GRIN2B
GO-MF	metal ion transmembrane transporter activity	57	4.73	8.3773 × 10^−8^	SLC13A2, SLC11A1, ATP1A2, SLC9A3, ATP2B3, TRPC7, SLC4A4, KCNK2, SLC4A11, SLC17A7, GRIN2D, GRIK5, CLDN16, SLC9A2, SLC5A7, SLC8A2, TRPM6, SLC17A1, SLC6A6, SNAP25, SCN7A, CACNG8, KCNN3, SLC4A10, KCNMB1, SLC17A4, SCN2B, GPM6A, KCNA6, SCN3A, KCNJ16, KCNMA1, SLC13A3, KCNB1, SLC5A11, GRIK3, SLC30A8, PKD1L2, SCNN1G, TRPV3, SCN11A, SCNN1B, SCN9A, SLC23A1, KCNG3, TMEM37, KCNK3, SCN4B, KCNA3, SLC17A8, SLC9A9, GRIN2A, KCNH8, KCNIP4, SLC30A10, SLC6A14, GRIN2B
	monoatomic ion channel activity	57	4.73	1.48449 × 10^−7^	CLCA1, CLCA4, BEST2, TRPC7, CNGB1, KCNK2, SLC4A11, GABRE, SLC17A7, GRIN2D, GRIK5, P2RX1, CLIC5, GABRG2, TRPM6, BEST3, SNAP25, SCN7A, CACNG8, BEST4, KCNN3, CNGA3, KCNMB1, SCN2B, GPM6A, KCNA6, GRIA4, SCN3A, KCNJ16, KCNMA1, KCNB1, GRIK3, PKD1L2, HTR3A, SCNN1G, TRPV3, SCN11A, SCNN1B, SCN9A, LRRC8E, KCNG3, TMEM37, KCNK3, ANO5, SLC26A9, SCN4B, KCNA3, SLC17A8, OTOP2, GRIN2A, KCNH8, CLCNKB, KCNIP4, HTR3E, GABRD, P2RX2, GRIN2B
	glycosaminoglycan binding	41	3.41	1.40577 × 10^−8^	ANOS1, COL11A1, CCN5, FGFR2, EPYC, SERPIND1, CTSG, BMP7, CEMIP, SFRP1, CCN4, COMP, PTN, CCL8, CCN6, JCHAIN, LYVE1, STAB2, HAPLN1, RSPO2, HABP2, ACAN, PCOLCE2, CLEC3B, SHH, TNXB, RSPO1, CXCL11, CXCL8, CEL, REG3A, ZG16, F2, GREM2, BGN, SLIT3, PRELP, SPOCK3, ELANE, VIT, CCL23
	extracellular matrix structural constituent	37	3.07	9.57517 × 10^−11^	ANOS1, COL11A1, COL19A1, LAMA1, SRPX, SRPX2, COMP, OGN, COL1A1, COL7A1, MFAP2, TGFBI, COL10A1, AMELX, FGL2, MATN3, CHI3L1, CILP, MMRN1, DPT, HAPLN1, HMCN2, ABI3BP, ACAN, EDIL3, CTHRC1, MFAP4, TNXB, FGB, COL6A5, BGN, EMILIN3, MUC6, PRELP, COL4A6, MFAP5, MUC5AC
	serine-type peptidase activity	36	2.99	1.1059 × 10^−8^	PRSS22, TLL1, FAP, CMA1, PCSK5, MMP11, CTSG, PRSS33, TPSG1, MMP8, PLAU, PCSK2, MASP1, KLK10, KLK8, DPP6, MMP7, MMP13, TMPRSS13, CORIN, HABP2, MMP3, TMPRSS3, MMP10, TMPRSS5, KLK6, KLK7, PCSK9, F2, RELN, MMP1, ELANE, CFD, PRSS41, PRSS56, PRSS2
	serine hydrolase activity	36	2.99	1.79471 × 10^−8^	PRSS22, TLL1, FAP, CMA1, PCSK5, MMP11, CTSG, PRSS33, TPSG1, MMP8, PLAU, PCSK2, MASP1, KLK10, KLK8, DPP6, MMP7, MMP13, TMPRSS13, CORIN, HABP2, MMP3, TMPRSS3, MMP10, TMPRSS5, KLK6, KLK7, PCSK9, F2, RELN, MMP1, ELANE, CFD, PRSS41, PRSS56, PRSS2
	serine-type endopeptidase activity	34	2.82	1.19537 × 10^−8^	PRSS22, TLL1, FAP, CMA1, PCSK5, MMP11, CTSG, PRSS33, TPSG1, MMP8, PLAU, PCSK2, MASP1, KLK10, KLK8, MMP7, MMP13, TMPRSS13, CORIN, HABP2, MMP3, TMPRSS3, MMP10, TMPRSS5, KLK6, KLK7, PCSK9, F2, MMP1, ELANE, CFD, PRSS41, PRSS56, PRSS2
	metallopeptidase activity	32	2.66	1.54386 × 10^−7^	PDPEP1, CLCA1, CLCA4, TLL1, ADAMTS6, TRHDE, ADAMTS2, MMP11, MEP1A, PAPPA2, MMP8, CPXM2, XPNPEP2, CPA4, ADAMDEC1, CPM, MMP7, MMP13, MEP1B, ADAM12, ADAM33, MMP3, ADAMTS12, CPB1, MMP10, ANPEP, NAALADL1, KLK7, LVRN, MMP1, MMP28, PRSS2
KEGG	Neuroactive ligand-receptor interaction	61	3.87	1.61 × 10^−11^	CHRM2, OXTR, THRB, NPFFR1, GRIK5, CHRM4, GRIK3, PTH1R, HTR4, ADRA1A, GHR, HTR7, UCN2, ADORA3, CTSG, PRSS2, INSL5, LYNX1, EDN2, AVPR1B, EDN3, GLP2R, NPY1R, TACR2, SSTR2, F2, SSTR3, GABRG2, SSTR5, ADRB3, AGTR1, NPSR1, CALCA, ADCYAP1R1, GRP, LPAR1, GRIN2A, APELA, P2RY4, CNR2, CNR1, NPY, P2RY1, PENK, GABRE, GABRD, DRD5, GRIA4, P2RY14, HTR1D, SCTR, GCG, GRIN2B, APLN, GRIN2D, KISS1, PYY, P2RX2, SST, P2RX1, VIP
	Cytokine-cytokine receptor interaction	41	2.60	7.87972 × 10^−6^	CCL13, CNTFR, CSF3, CSF2, CXCL8, TNFRSF13B, IL24, CXCR5, CXCL17, CXCL1, CXCL3, CXCL2, CXCL5, GHR, CCL8, TNFRSF17, CCL19, AMH, IL6R, CCR2, IL11, CCL23, TNFRSF12A, CCL21, GDF15, OSM, LIFR, PPBP, INHBA, BMP7, BMP5, INHBE, EDAR, BMP3, IL1A, CXCL11, CXCL12, IL23A, TNFSF9, CCL28, IL17C
	cAMP signaling pathway	35	2.22	3.09924 × 10^−6^	CHRM2, OXTR, ADCYAP1R1, HHIP, ATP1A2, HTR4, ADCY5, GRIN2A, PLN, CREB3L3, NPY, PLCE1, TNNI3, CNGA3, AMH, BVES, PRKACB, GRIA4, DRD5, EDN2, EDN3, HTR1D, NPY1R, ATP2B3, GCG, ATP1B2, SSTR2, GRIN2B, GRIN2D, SSTR5, SST, VIP, KCNK2, MYL9, CNGB1
	Calcium signaling pathway	31	1.97	0.000983789	CHRM2, OXTR, HTR4, ADRA1A, SLC8A2, MYLK, GRIN2A, HTR7, PLN, FGF20, PLCE1, NOS1, PRKACB, DRD5, PRKCG, AVPR1B, PRKCB, TACR2, ATP2B3, VEGFD, GRIN2B, GRIN2D, ADRB3, P2RX2, P2RX1, FGF19, CASQ2, AGTR1, PLCD4, PLCD1, FGFR2
	Cell adhesion molecules	25	1.59	8.07266 × 10^−5^	NLGN1, NRXN1, VTCN1, CLDN2, CLDN1, CDH3, SLITRK2, MPZ, SLITRK3, CLDN23, NCAM1, MADCAM1, JAM2, NTNG1, CADM3, NEGR1, CLDN11, IGSF11, CLDN14, CLDN8, ITGA8, CNTN1, CNTN2, CLDN16, CD22

**Table 3 ijms-25-03954-t003:** The enrichment status of 47 hub genes in the (GO) analysis.

	Name	Genes
GO-BP	production of molecular mediator of immune response	IGKV2-24, IGKV2D-29, IGLV7-43, IGLV8-61
	immunoglobulin production	IGKV2-24, IGKV2D-29, IGLV7-43, IGLV8-61
GO-CC	immunoglobulin complex	IGKV2-24, IGKV2D-29, IGLV7-43, IGLV8-61
	collagen-containing extracellular matrix	PRELP
	apical plasma membrane	CA4
	cell projection membrane	CA4, GABRE, NME1, TSPEAR, ITLN1
	monoatomic ion channel complex	GABRD, GABRE, GRIK5
GO-MF	metal ion transmembrane transporter activity	GRIK5
	monoatomic ion channel activity	GABRD, GABRE, GRIK5, CLCA1
	glycosaminoglycan binding	PRELP
	extracellular matrix structural constituent	PRELP
	serine-type peptidase activity	MMP1, MMP3, TPSG1
	serine hydrolase activity	MMP1, MMP3, TPSG1
	serine-type endopeptidase activity	MMP1, MMP3, TPSG1
	metallopeptidase activity	MMP1, MMP3, CLCA1
IGKV2-24, IGKV2D-29, IGLV7-43, IGLV8-61, PRELP, CA4, GABRE, NME1, TSPEAR, ITLN1, GABRD, GRIK5, CLCA1, MMP1, MMP3, TPSG1 (Summarize: A total of 16 genes)		

## Data Availability

All data sets used in this study are publicly available on the TCGA.

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
