# Peer review of "The Construction of a Multi-Gene Risk Model for Colon Cancer Prognosis and Drug Treatments Prediction"

_ijms, 2024, doi:10.3390/ijms25073954_

Round 1

Reviewer 1 Report

Comments and Suggestions for Authors

The author should provide a weblink for TCGA COAD. After searching TCGA COAD, I found the web portal at https://portal.gdc.cancer.gov/projects/TCGA-COAD, which mentions 461 samples. The author should include the web portal link and provide information about the mRNA expression profiles of the 471 Colon Adenocarcinoma (COAD) samples downloaded for this study.

Many studies and analyses of COAD are mentioned in cBioPortal. The author should elaborate on what readers can gain from this analysis conducted in this study in comparison to others available on cBioPortal. Clarifying the unique contributions and insights provided by the current analysis will enhance the overall understanding of the study's significance.

The author should explain the results of the survival curves, emphasizing which genes are more significant in their survival analysis. Additionally, it would be beneficial to clarify why the author did not choose to compare the survival analysis of these genes across all clinical stages. Such analyses are often help to define prognostic indices for mortality or disease recurrence and to assess treatment outcomes.

 The author should summarize the significance of the three R package analyses conducted in this study. Which is more significant .This will assist readers in understanding the key insights gained from these analyses.

Reviewer 2 Report

Comments and Suggestions for Authors

The manuscript Potential of multi-gene isolated against colon cancer was reasonable and technically sound. 

Below are some suggestions.

Point 1Line 14, 20, 23… : Check the spacing throughout.

Point 2. Please be more specific in the results section.

Point 3. Certain(CDKN2A, CXCL1, CLCA1...) genes have already been studied in colon cancer. Please suggest the possibility that the 27 genes are novel or linked to each other and can be used for treatment or prediction.

Comments on the Quality of English Language

Minor editing of English language required

Round 2

Reviewer 1 Report

Comments and Suggestions for Authors

The authors' responses to be comprehensive and convincing, and I agree to accept the revised manuscript. Congratulations to the authors!